# Services and Amenities Offered by City Hotels within Family Tourism as One of the Factors Guaranteeing Satisfactory Leisure Time

**DOI:** 10.3390/ijerph19148321

**Published:** 2022-07-07

**Authors:** Matylda Siwek, Anna Kolasińska, Krzysztof Wrześniewski, Magdalena Zmuda Palka

**Affiliations:** 1Department of Tourism and Regional Studies, Institute of Geography, Pedagogical University, 30-084 Krakow, Poland; anna.kolasinska@up.krakow.pl; 2Section of Psychology, Institute of Humanities, University of Physical Education, 31-571 Krakow, Poland; krzysztof.wrzesniewski@awf.krakow.pl; 3Section of Pedagogy, Institute of Humanities, University of Physical Education, 31-571 Krakow, Poland; magda-zmuda@wp.pl

**Keywords:** hotels, family tourism, facilities and amenities, leisure time, children, kid-friendly

## Abstract

Travelling families are interested in urban tourism due to its cognitive and entertainment aspects. Some expectations of parents travelling with children are the same independent of the accommodation location. The purpose of this article is to examine at what level city hotels offer services and amenities within family tourism, how they meet parents’ expectations and whether they adapt offers to children of different ages. Two measurement tools were used for this study. The first one was the authors’ questionnaire addressed to hotel managers. A total of 88 hotels located in Krakow were selected for the study. The survey contained questions on the offers of hotels related to various services and amenities for families with children, which were assigned to one of three groups (24 items): general child-friendly hotel services and amenities, in-room amenities and restaurant amenities. In the second phase of the study, parents of children aged 0–10 years were asked to assess the degree to which the amenities offered by hotels were important for them during the selection of a hotel. In total, 90 surveys were taken into consideration. The tool was designed to examine the needs of parents who have children of different ages, in regard to selected hotel services and amenities (24 items). As in the case of the surveys for hoteliers, the same three groups were created. The research demonstrates that children’s age is the key factor that should lead to the differentiation of the accommodation offer. However, hoteliers largely perceive children as a homogeneous group. In all studied age categories (6) and amenities groups (3), parents’ expectations were higher than the services and amenities offered. No significant differences regarding the category of the hotel or hotel chain membership have been found. The results of the study may provide valuable guidance to hoteliers who wish to attract families with children to urban tourism and adapt their offer to them, thus realising the principle of inclusiveness. They may be also a significant contribution to the literature on the subject, as most published work concerns holiday hotels rather than city hotels.

## 1. Introduction

One of the most distinctive features of tourism as an industry is its dynamism and ability to adapt to changing conditions and the needs of consumers—in this case, tourists. The mechanism mentioned above also applies to family tourism, which, given the above dependence, is developing vigorously. Families with children are significant shareholders in the tourist and hotel industry, and their needs must therefore be taken into consideration by tour operators [1,2,3]. In 2012, Schänzel et al. [4] noted that tourism and hotel industry operators who neglected to prepare an appropriate and satisfactory offer for families with children would put themselves at a disadvantage in the market. In view of the growing role of family tourism and the fact that children are important customers in the hospitality industry, it has become necessary to adapt the accommodation base to their needs [5,6]. Deficiencies in this respect may be the cause of great stress instead of leisure during a family holiday [7]. Accommodation plays an important—if not the most important—role during a family holiday, and younger family members often impact the decisions taken by families in this respect due to their specific requirements [8].

The organisation and service of family tourism should take into account the changing structure of the family, the increase in the number of multi-generational travels and the changing objectives and priorities of family holidays [4]. The reasons for family holidays include leisure, but also education, entertainment, excitement, the desire to experience an adventure and the acquisition of new experiences and skills [9,10].

Studies in the field of family tourism most often relate to holidays. Research on this topic includes the decision-making processes [11,12,13,14] or behaviours of families as consumers [15,16,17]. the decision-making difficulties in multi-generation families in regard to holidays have been highighted. Research into the functioning of family tourism in hotels is largely focused on selective services such as mini-clubs [18], leisure offers and animation programs [18,19,20], childcare [21] and hotel facilities. Such research mostly concerns resort hotels [6,22,23,24]. Work has focused on the important elements of the infrastructure designed for the youngest tourists (e.g., access ramps for baby carriages, leisure area, children-only toilets), which made it easier to stay in a touristic destination. In [5], the authors proposed an interesting approach, focusing on family hotels’ digital communication strategies; they also explored the type of messages hotels convey on digital platforms and how they redesign their services to attract families with children. Studies were also carried out, albeit to a lesser extent, at hotels in large cities, for example in Paris [25], and they focused on the participation of families with children in urban tourism. In the hospitality and catering industry, holiday hotels are the main places preparing offers designed for families with children. They know how to create an interesting offer and attract the above-mentioned tourist group [5]. However, the opposite situation has been noted in the case of city hotels. According to a report published in the French “Le Progres” journal at the end of 2018, only 5% of hotels in Paris were child-friendly [5], most of which offered small rooms, so parents often chose to rent two separate rooms. In [26], the authors suggested that cities were not suitable destinations for family travels. However, due to the increasing educational and cognitive—among others—roles of family holidays, this type of tourism could also be attractive to families with children.

An in-depth analysis of the reference literature has revealed that little research has been focused on the preparation of hotels for family tourism in a comprehensive manner. This would include the analysis of both the demand and the supply sides. The vast majority of works relate to holiday tourism, holiday hotels or camping sites. There are also studies on families with children in urban areas that have been undertaken, e.g., on visiting museums [27] or eating in restaurants [2,28], but there is a shortage of studies discussing children’s participation in urban tourism, including the preparation of hotels to accommodate such guests with their specific requirements. This paper attempts to fill in this gap. It proposes the division of the services and amenities offered by city hotels into three groups. The amenities and services offered in the common space of the hotel (general child-friendly hotel services and amenities), rooms (on-room amenities) and restaurants (restaurant amenities) are discussed separately. Moreover, the vast majority of family tourism research refers to the demand side, i.e., the needs, preferences and likings of tourists. This article combines research both on the demand and supply sides, combining the services and amenities offered by hotels with the expectations of the parents. Given the fact that all the accommodation facilities in question are located in the touristic centre of one of Europe’s most attractive cities, special attention has been paid to the specific characteristics of urban tourism. Krakow’s attractiveness and the value of its national heritage has already been recognised, and it was inscribed onto the UNESCO World Heritage List in 1979.

The aim of the study is to research the services and amenities of city hotels directed at families with children and to examine how city hotels meet the expectations of parents with young children (0 to 10 years old). The following questions have been raised: can significant differences in offered amenities depending on hotel category and network membership be observed? Which amenities do hotels offer most often? Which amenities are the most important for parents? Do the parents’ needs for specific amenities depend on the age group of their children? To what extent do the services and amenities offered by hotels meet the parents’ expectations?

From a manager’s point of view, this article can help hotel managers and hospitality professionals to understand the different needs of families with children to properly redesign the services and amenities offered.

The article contains six sections, beginning with the Introduction and ending with the Conclusions and Limitations. The literature review describes how family tourism issues, particularly the adaption of hotels to the needs of families with children, are viewed in the literature. Then, Materials and Methods, Results, and a Discussion are presented.

## 2. Literature Review

### 2.1. Family Tourism—Main Issues

Many of the trends concerning family tourism observed in previous years are still valid. Peter C. Yesawich stated in 2007 that family tourism would develop at a faster pace than other forms of leisure travel, because it was perceived by parents and grandparents as a means of family reunification [29]. This is an extremely important aspect, especially today, when daily duties consume huge amount of work and time. In a study carried out in 2019 by Trafalgar Tours on a sample of 6000 families, 64% of parents admitted that the best quality time with their children was during their travels [30]. Tourism has been playing an important role in building family ties [7,31,32,33,34,35], and the experience of travelling together has a positive impact on the functioning of the family [13,36,37]. Sometimes the only possibility for family members to spend more time with one another is at the weekends, during public holidays and vacation. Tourism has become one of the main ways that families spend their free time together, and parents are willing to spend increasing amounts of money and energy on travelling with children [38]. Families have been looking for destinations that offer relaxation, the opportunity to spend their time outdoors, discover art and cultural heritage, but also include adventure, excitement and some form of “newness” [34].

The diversity of families should be taken into account during the creation of touristic services for families. Most commonly, families are described in the literature as two heterosexual parents with a child or children. By contrast, this term should also mean single parents with children, parents of the same sex or mixed families (e.g., new parents’ families after a divorce) [34]. Cultural differences, different values embodied in families and the objectives that are to be achieved through travels as well as socio-economic aspects should also be taken into consideration [39]. Furthermore, children’s different age categories (from infants to young people) should also be considered during the preparation of an offer for tourists, both in terms of services and infrastructure [23]. Children have other needs and requirements [38] that the tourism industry should meet in order to make travelling a happy and satisfying experience both for them and their parents. In [21], the authors noted that families with children of different ages were the most difficult in regard to meeting various expectations and requirements.

Another specific feature of family tourism is the fact that the motivational and decisive factors of the nature of travel depend not only on the adult family members but also—or even above all—on the children. Family tourism research has often been concentrated on parents [36,40]; however, the literature also shows studies of the perception of family holidays with the eyes of children [10,23,41,42,43,44] or teenagers [39]. Parents’ decisions related to the choice of a place, accommodation or a type of holiday are guided by the preferences of their children, following the principle that “a happy child is a happy parent”. The more interesting (often remarkable) proposals for children that there are and the more amenities in the hotel facility there are, the more parents can rest, relax and enjoy their holiday in a tranquil manner [36]. At the same time, the above-mentioned aspect of family tourism, i.e., the possibility of spending time together and strengthening family ties, will be fulfilled.

This trend is linked to the reassessment of the child’s role. It is children who are becoming the main consumers of family tourism, and they are the target of touristic offers, which take into consideration their needs and preferences. In [2], the authors stressed that children were extremely important for the hotel sector and should be seen as active decision-makers because their satisfaction affected the satisfaction of their parents. Hoteliers see their future potential customers in today’s children, as preferences acquired during childhood are frequently maintained in adult life [45]. The need to focus on the greater role of children in the decision-making process and to treat them and—and not the parents—as the main decision-makers was already emphasised in the 1990s [41]. This aspect has also been taken into account in the following years [2,19]. The authors of [2] clearly stated that it was extremely important to identify, recognise and meet the needs of the youngest participants of travels, as this contributed to creating positive emotions that encouraged and enhanced consumer loyalty and the positive message. Moreover, according to Curtale’s studies [46], children’s preferences affect parents’ choices, with parents willing to pay more to meet their children’s preferences, and parents’ willingness to pay may vary depending on the response of their children. Ceylan (as cited in [47]) also stated that children were the most efficient family members when choosing a holiday destination. According to her, parents first consider the preferences of their children, regardless of their age. The literature emphasises the importance of amenities in the context of the expectations of parents, grandparents and children. However, there is only scant research comparing them with the hotel offer, namely on the supply side.

### 2.2. Urban Tourism in the Scope of Interest of Family Tourism

Since ancient times, cities have attracted many visitors [48]. The phenomenon of urban tourism lies, among other things, in the fact that cities have a huge variety of possibilities for tourists, and they offer countless attractions. Cities allow one to relax, spend time with family or friends away from everyday problems and responsibilities and provide a huge aesthetic, cognitive and educational experience at the same time. They provide people with the opportunity to interact with the whole diversity of culture and art that has been being created for centuries or even thousands of years. People can take part in a variety of events, increase their level of knowledge, acquire new experiences and skills and even interact with nature when visiting gardens, such as botanical or zoological gardens, parks or nature reserves, which are often located on the outskirts [48,49,50]. There are various types of tourism that can develop in the urban area. These can be connected with business, educational, cultural, cognitive, medical, sports or entertainment reasons [51,52]. There are also different segments of urban tourism recipients, such as adults, young people, children coming for school trips, families with children, seniors, people with disabilities, businessmen, etc. [49,53]. Infrastructure development and adaptation is a must to meet the needs and expectations of such diverse groups of visitors.

Holiday tourism is a dominant form of travel preferred and chosen by families, but due to the increasing emphasis on educational and cognitive aspects of travel, urban tourism is becoming more popular, also among families. Cultural institutions, museums and theme parks have been among the first to recognise children as important customers to whom an offer should be addressed [54]. In other authors’ studies, sightseeing is a frequently mentioned holiday activity. Respondents in a study [55] who were multi-family members ranked sightseeing fourth (89%) among their preferred activities. Further positions on the list were *going to museums/exhibitions* (45%), *visiting a theme park* (35%) and *going to the theatre*, *musicals and opera* (24%). Theme parks and cultural or famous sites are visited during travels to cities, which are the main travel destinations or an additional activity during resort holidays [23]. The experiences that even the youngest children acquire during visits to museums, science centres, etc., shape their perception of the world, their sensitivity and their willingness to acquire and expand knowledge in later years. They develop their imagination. They are also great spaces for families to spend their time together and thus strengthen their family ties [27,56,57,58]. It is important that the above-mentioned amenities are adapted to the needs of children [56] and parents at the same time [59]. The museum space allows parents to have their own leisure time, while children participate in activities such as workshops [60].

### 2.3. Adapting Hotels to the Needs of Families with Children

During holidays, families often choose to stay in hotels [55], which is why it is so important to adapt offers, infrastructure and amenities to their needs. Travelling families are not typical tourists. The needs and consumer behaviours resulting from them are different from those of other tourists [16,61]. In [62], the authors noted that package offers targeted at families focused more on the interaction of children with other family members, and the emphasis on consumer benefits seemed to be stronger in comparison to other touristic offers. Families with children also make a different choice of accommodation [11,63], above all positively assessing those facilities which provide amenities and food for children [6]. Furthermore, in [20], it was stressed that a comprehensive approach was needed when designing amenities and services for young children. Accessibility, the possibility of interacting with other children, safety, hygiene, room size and animation programs for children should all be taken into consideration. The need to introduce some amenities for young tourists, particularly the organisation of childcare and attractive animation programs, was highlighted in the early 1990s. Even then, many hotels and holiday centres realised that the development of special programs for children could contribute to the success of a hotel [64]. However, the global crisis caused by the COVID-19 pandemic has made hotel guests not only pay attention to hygiene amenities (e.g., disinfectants) but even to consider them as among the most important factors [65]. As Kim and Han [65] suggested, there has also been an increased interest among visitors in services without human interaction, often based on the latest contactless technologies. Similar conclusions in terms of the introduction of innovative technologies for various facilities were reached by Cheung et al. [66]. Buzlu and Balik [67] found that it is important to have a children’s tab on the hotel’s website to provide parents with easy access to information about the hotel’s child-related facilities.

The main players in preparing a child-friendly offer in the hospitality industry are resorts, which, as pioneers in this field, know how to attract and provide the right conditions for families to spend time at their facilities. The opposite is true for city hotels, which only now are starting to perceive families with children as their potential customers [5].

#### 2.3.1. Family Rooms and Equipment Elements

A lack of sufficient space during a family holiday may cause conflicts between family members, as noted by [7,68]. The provision of large and spacious accommodation units for families is therefore the basis for the service of this type of guests. Carr [69] suggested that accommodation providers who promoted their facilities as family-friendly in reality offered only a little bit more than cramped family rooms, but in reality, they rather focused on providing a variety of activities targeted at children that took place outside the bedrooms, in places such as children’s clubs or swimming pools. Carr [69] also pointed out that there was an increasing number of hotels that were trying to meet the individual needs of children and their parents by designing family-friendly rooms that provided a certain degree of privacy.

#### 2.3.2. Play Rooms, Mini Clubs

As shown above, for a family travelling with children, it is important to know if there is a play area in the common space, such as a play room or a mini club space, when taking a decision on accommodation facility. The play room is understood as a separate space for a child equipped with various equipment and amenities, i.e., jungle gyms, small trampolines, tents or villas, tunnels, table football game, ping-pong table, etc. In contrast, mini clubs are much more likely to be found in holiday hotels, and they offer a variety of activities for children, led by qualified personnel. Regardless of the location of the hotel, certain activities and programs provided by mini clubs to make the stay at the hotel more attractive are often similar [19,70,71]. Research by [3] largely focused on the benefits of mini clubs as a tool with the potential to create a competitive advantage. The authors pointed out that the activities proposed in such places, based on unique local themes (heritage, culture) and original materials, as well as the involvement of the local community and building a positive image of the centre, might become key differentiating elements for family hotels. In [72,73], the authors also emphasised the educational role of hotel animation programs, stressing that the idea of sustainable development could be implemented into the activities conducted for the youngest in the mini club on holiday. In addition, not only do young people see animation as a healthy or entertaining activity, but also as an opportunity to learn more about their peers from different countries [74]. However, in [21], the authors stressed that families from different cultural circles might have different expectations regarding the purpose and scale of activities organised for children in accommodation facilities. Research among parents from Asian countries showed that they expected more educational activities on holiday [6], while parents from Western countries (“Western”) would put more emphasis on their unique character and originality (novelty and out-of-the-box) [34]. The diversity of expectations poses challenges for accommodation facilities to prepare original and at the same time universal offers. Families with children in particular want to choose hotels that are aimed both at themselves and at their children [47]. The inclusion of activities aimed at whole families, not just children, in the hotels’ offerings can be an important element of becoming a family-friendly hotel [67].

#### 2.3.3. Restaurant Amenities

An important element of staying in a hotel is eating meals, which is seen not only as a necessity and a way of meeting physiological needs but also as a pleasure. As noted by [75], food is linked to the maintenance of family relationships that intertwine with memories and help to develop family cohesion. That is why not only accommodation services but also food services are so important in hotel facilities. Tourists—in this case, families with children—pay attention not only to the place where the meals are eaten (atmosphere, cleanliness, service) and how tasty and healthy the meals are but also to the way they are served, what kind of a menu offer there is for children, the hourly availability and how the table is laid. It is also important to have equipment that makes it easier to eat a meal with a small child, such as a special chair or children’s area in which they can spend their time waiting for a meal. Parents also appreciate the possibility of heating ready meals for young children or ordering meals for children with different nutritional requirements, such as food for allergy sufferers or those intolerant to gluten. The study by Hay [43] showed that children were critical towards some aspects of food in hotels, such as menus that were difficult to understand as well as excessive portions. In the study on Chinese children’s touristic experience, it was noted that food was becoming a special attraction itself in many places, as it could provide children with a variety of sensory and cultural experiences, especially when they tasted local dishes [44].

## 3. Materials and Methods

### 3.1. Research Design—Participants and Research Tools

Two measurement tools were used for this study. The first research tool was the authors’ questionnaire addressed to hotel managers located in Krakow in the areas of the Old Town and Kazimierz, which are two touristic districts. Krakow is the most recognised Polish city in the world—14 million tourists visited it in 2019 before the pandemic [76]. In terms of the number of hotels, Krakow is in first place in Poland with its 173 hotels and 11.6 thousand rooms [77]. The hotels to be examined were selected on the basis of the Central List of Hotel Facilities in Poland, which collects data about all objects subject to categorisation in accordance with the Polish law [78]. A total of 88 hotels were selected for the study, 19 of which refused to participate in the research.

In terms of hotel development, Krakow is one of the leading cities in Poland. The market is dominated by three-star hotels, which account for the largest share (51%) of the total number of such objects in the city, while four-star hotels have the largest share in terms of the number of hotel rooms. Overall, there are approximately 21,000 beds in Krakow’s hotels. A characteristic feature of Krakow’s hotel industry is small hotels, often located in historic tenement houses, some of which date back to the 19th century. Most hotels in Krakow are either independent or belong to Polish hotel chains, although for the last dozen or so years, the city has also attracted the interest of international chains (Accor, Hilton, IHG Hotels & Resorts, Marriott International, Radisson Hotel Group), whose hotels are established in various parts of Krakow. There are currently 32 hotels operating in Krakow under international brands. The paper-based survey questionnaires were provided to each of the hotels chosen at an earlier stage, together with a letter of intent informing about the purpose and scope of the research project as well as researchers’ contact details. The surveys were filled in by the owners, directors and managers. No incentive was provided with this study. The study was conducted from November 2019 to January 2020.

The decision on the structure of survey questions was taken after an in-depth study of literature related to the research connected with the hospitality and family tourism topics. Many of the topics, which related to elements specific for a typical holiday hotel (such as swimming pools, mini-clubs and animation programs), had already been raised in previous studies. However, some amenities connected with services and equipment were important for families regardless of the type of a hotel or its location. The survey contained questions on the offers of hotels related to various services and amenities for families with children, which were assigned to one of three groups (24 items):(1)General child-friendly hotel services and amenities—doctors’ phone numbers; discount vouchers for children’s attractions; adequately wide lifts for baby carriages; a place to store baby carriages; access ramps; play area in the lobby; play room for children; gifts for children; outdoor playground; honouring Large Family Cards; special packages and discounts for families with children.(2)In-room amenities—baby cots; extra beds for older children; stools for children; heaters; television with programmes for children; baby bathtubs; security (e.g., plugs, table corners, windows).(3)Restaurant amenities—children’s menu; high stools for children; children’s tableware; play area; toilet with baby diaper changing tables; a room for a mother with a baby; separate toilet for children.

In the second phase of the study, parents of children aged 0–10 years were asked to assess the degree to which the amenities offered by hotels were important for them during the selection of a hotel. The information part clearly indicated the required age of children and that the questions concerned the services and amenities offered by city hotels. The survey questionnaire was sent to parents who declared that they travel at least for a single night trip once a year. The survey was published electronically on various online forums for parents and sent to parents whose children attended randomly selected schools and kindergartens. In total, 91 responses were received; due to the non-fulfilment of the formal criterion of one of the parents (age of children over 10 years), 90 surveys were taken into consideration for further analysis. The study was conducted in February 2021. There were no incentives in this case as well.

Respondents were treated as one group and additionally were divided into six groups: (1) parents of children aged 0–3 years; (2) parents of children aged 4–6 years; (3) parents of children aged 7–10; (4) parents of two or more children aged 0–6 years (e.g., first child aged 2 years and second child aged 5 years); (5) for parents of two or more children aged 4–10 years (e.g., first child aged 4 years and second child aged 8 years); (6) parents of three or more children aged 0–10 years (e.g., first child aged 3 months, second child aged 4 years and third child aged 9 years).

The tool was designed to examine the needs of parents who had children of different ages, in regard to selected hotel services and amenities (24 items). As in the case of the surveys for hoteliers, the same three groups were created: general child-friendly hotel services and amenities, in-room amenities and restaurant amenities.

Parents responded to the five-point Likert scale. The obtained results were used to build a ranking of amenities; the arithmetic mean and standard deviation were calculated for all hotel amenities. Then, each amenity received between 1 and 5 points: 1 point was allocated to amenities with an average value less than the average value of all amenities measured together (M_a_) – the standard deviation of this value (SD_a_); 2 points were allocated to amenities with an average value within the following range: (M_a_ – SD_a_; M_a_ − 0.5*SD_a_); 3 points were allocated to amenities which were in the range of M_a_ ± 0.5*SD_a_; 4 points were awarded when the result was in the following range: (M_A_ + 0.5*SD_a_; M_a_ + SD_a_); and 5 points were awarded when the result was >M_a_ + SD_a_. The next step was to calculate the average ranking value for each amenities group.

In this way, seven rankings were prepared: (1) for parents of children aged 0–3 years old; (2) for parents of children aged 4–6 years old; (3) for parents of children aged 7–10 years old; (4) for parents of two or more children aged 0–6 years old; (5) for parents of two or more children aged 4–10 years old; (6) for parents of two or more children aged 0–10 years; and (7) for all parents.

After collecting information about the facilities of hotels, the value of the hotel was calculated for the three groups of amenities. The result was calculated taking into account whether an amenity (the same as that included in the survey for parents) was offered at the hotel. If not, the hotel received 0 points. If yes, the hotel scored as many points as there were allocated to a certain parent group. Then, an average point value for a hotel was calculated for each group of amenities, taking into consideration the offer that was aimed at parents with children of different ages.

This approach is different from the existing studies because it tries to investigate the inclusiveness of hotels for family tourism, both from the perspective of the hotel and expectations of parents with young children. It is in line with the postulates of Dowse et al. [1] and Radic [79], who argued that in order to get to know and understand the role of children as an important group of consumers in the hotel sector, there is a need to undertake various studies.

### 3.2. Data Analysis

The Shapiro–Wilk test was used to examine whether the distribution of the variables studied was close to a normal distribution. Based on the results of this test, there was no indication that most of the examined variables had a distribution that was close to normal. Due to the above and the number of the surveyed groups, it was decided to use non-parametric tests.

The Kruskal–Wallis test was used to compare the hotel services and amenities offered to families with children, taking into consideration the category of a hotel, while the Mann–Whitney *U* test was used to compare them according to their membership in the hotel network.

The Friedman test with dependent variables was used to investigate which of the most frequently proposed amenities were offered by hotels; the resulting rankings (for each age group separately) of the three groups of amenities described above were compared to each other (three dependent variables). The Friedman test is the non-parametric equivalent of the one-way analysis of variance for dependent variables [80]. Therefore, it allowed us to investigate whether there are significant differences between the rankings of amenities offered by Krakow hotels and which of these amenities best meet the expectations of parents of children of different ages.

The Mann–Whitney U test was used to compare parents’ expectations regarding hotel services and amenities with the existing offers of the hotels.

Missing data treatment: there were no missing data in either the hotel survey or the parent survey.

## 4. Results

The Kruskal–Wallis test did not indicate statistically significant differences in hotel offers in different categories (*p* > 0.05). Furthermore, the Mann–Whitney *U* test did not indicate any significant differences between hotels belonging to a hotel chain and independent hotels (*p* > 0.05).

### 4.1. Amenities and Services for Families with Children Offered by Hotels

The results obtained in the hotel surveys are summarised in the table below (Table 1), showing the percentage of hotel facilities with the amenities and services offered by the hotels. In addition, they are compared with parents’ expectations (calculated in total, with no distinction between age categories; points are the average of the responses (rounded to the nearest whole number) regarding the expectations for a particular amenity).

### 4.2. The Most Important Amenities Groups Offered by City Hotels for Parents

The average ranks for the three groups of amenities were calculated (general child-friendly hotel services and amenities, in-room amenities, restaurant amenities) on the basis of the results of the parental survey. The results obtained are shown in Table 2. It can be concluded that for all the parents, the most important amenities are general child-friendly hotel services and amenities, a little less important are restaurant amenities (0.3 points less (8.4%) compared to the first mentioned group of amenities), while in-room amenities get the lowest scores (by 0.9 points less (25.1%)). Similar results can be observed for parents whose children are 4–6 years, 7–10 years, 0–6 years, 4–10 years and 0–10 years of age. The only exception is the group of parents with children aged 0–3. It has been noted that this age category has a similar demand for amenities as all three groups: restaurant amenities have been defined as the highest priority, followed by general child-friendly hotel services and amenities and finally by amenities from the in-room amenities group (Table 1).

### 4.3. Amenities for Families with Children Offered by City Hotels in Comparison with Parents’ Expectations

The results obtained from the surveys conducted at hotels were compiled together with the surveys conducted among parents. Average score ranks were created. They show points obtained by the city hotels that took part in the survey based on scores calculated on the basis of parents’ expectations in general and in one of the age categories (the procedure for calculating the results is presented in Section 3.1). The results are presented in Table 3.

On the basis of the results of the Friedman test, statistically significant differences in parents’ expectations have been identified, irrespective of their children’s age, taking into account the three amenities groups (*chi^2^*_(2, 69)_ = 39.095, *p* < 0.001). It has been noted that parents’ expectations are met to a higher degree with general child-friendly hotel services and amenities (M_rang_ = 2.55) than restaurant amenities (M_rang_ = 1.96). The expectations regarding in-room amenities (M_rang_ = 1.49) are least met. Regarding the results of the other groups of parents, the same differences have been noted in the group of parents of children aged 7–10 years (*chi^2^*_(2, 69)_ = 11.246, *p* = 0.004). The parents’ expectations are met to the highest degree in the case of general child-friendly hotel services and amenities (M_rang_ = 2.32), then restaurant amenities (M_rang_ = 1.91). The expectations regarding in-room amenities (M_rang_ = 1.77) are least met. In addition, significant differences have been noted in the offers addressed to families with many children aged 0–10 years (*chi^2^*_(2, 69)_ = 15.590, *p* < 0.001). The expectations regarding in-room amenities (M_rang_ = 1.62) are least met. The expectations related to the other two groups are met with similar results (M_rang_ = from 2.12 to 2.26).

No statistically significant differences have been identified (*p* > 0.05) in terms of meeting parents’ expectations for children aged 0–3 years, 4–6 years, 0–6 years and 4–10 years (Figure 1).

The Friedman test has demonstrated that there are statistically significant differences in amenities targeted at parents of children of different ages in relation to the general child-friendly hotel services and amenities group (*chi^2^*_(5, 69)_ = 105.284, *p* < 0.001). This offer is least in line with the expectations of the families with many children aged 0–10 years (e.g., first child aged 1, second child aged 5 and third aged 8) (M_rang_ = 1.70) and the families in which there is a child aged 4–6 years (M_rg_ = 3.17). The expectations of parents from other groups were met in a similar way (M_rang_ = from 4.03 to 4.05). Significant differences have also been noted for the in-room amenities group (*chi^2^*_(5, 69)_ = 272.576, *p* < 0.001). As with the previous group of amenities, the offers met the expectations of the families with many children aged 0–10 years (M_rang_ = 1.79) and of families with a child aged 7–10 years (M_rang_ = 2.17) and 4–6 years old (M_rang_ = 2.34) least. In the case of the restaurant amenities group, there are also statistically significant differences as regards the offer for families with children (*chi^2^*_(5, 69)_ = 90.569, *p* < 0.001); the hotel portfolio is the least suitable to the needs of parents of children aged 7–10 years (M_rang_ = 2.60) and 4–6 years (M_rang_ = 2.76). In the case of families with many children, the offer related to this amenities group met parents’ expectations in a similar way (M_rang_ = 3.32 to 3.89). The hotels’ offers have been mostly targeted at parents of children aged 0–3 years (M_rang_ = 5.11).

The results of the Mann–Whitney U test, carried out to examine to what extent the amenities offered by Krakow hotels meet parents’ expectations, show statistically significant differences between the services and amenities offered by city hotels to families with children (general child-friendly hotel services and amenities, in-room amenities and restaurant amenities) and the expectations of parents of children of all ages (*p* < 0.001).

Table 4 shows to what extent (presented in percentages) the hotel offers meet the parents’ expectations (the score of parents’ expectations was treated as 100%, then the percentage range in which the score of the hotel offer fell was measured). As regards the all parents group, hotels met expectations for the general child-friendly services and amenities group in 51.1% of cases, the in-room amenities group in 35.3% of cases and the restaurant amenities group in 35% of cases. By analysing the percentage values in regard to meeting the expectations of parents of children of different ages (Table 4), it can be observed that hotels only met their expectations to a small extent (24.7% to 32.7%).

## 5. Discussion

The main destinations for family holidays, especially during the holiday season, are holiday resorts, spa towns and seaside resorts. On the other hand, following the trend of combining travelling with educational objectives, cities may also attract families with children. The specificity of city hotels differs from holiday and spa hotels, as the main purpose of the guests is not to stay in the hotel and use its infrastructure but to explore the city and its attractions. Having said that, parents travelling with children have certain requirements and expectations regarding amenities and facilities at a hotel during such travels.

On the basis of the studies carried out, it has been demonstrated that neither the category of city hotels nor their membership in a hotel chain are indications of the existence of specific amenities for family tourists. This is a surprise because it could be expected that higher-ranking hotels, due to their higher standard, would offer a wider range of services, including those for families with children. Such conclusions were presented in [22] when examining coastal hotels in Turkey. The author stated that it was five-star hotels that were the most family-friendly. This study does not confirm that, which may result from the specific characteristics of city hotels, for which family amenities are still not a part of their competitive advantage. Another important element influencing the presented results may be the regulation in force in Poland as regards the categorisation of hotels. According to this, hotels do not have to meet certain requirements for family-friendly services. In [81], the authors demonstrated that five-star hotel facilities were not popular with families with children. It is also worth noting that the top-of-the-range hotels are expensive, and for many families with children, travel costs are still an important element in the holiday budget. That may be the reason why the top-of-the-range hotels do not perceive parents with children as their customers. Therefore, they do not see the need to meet their requirements, which always involves increased costs for the hotel [68].

As in the case of the categories, the membership of a city hotel in a chain of hotels does not guarantee that there are some amenities for families with children. Independent hotels, which are often smaller, do not see the need to increase their competitiveness as regards their offer for families with children. Specific hotel brands, which function as hotel chains, often prepare their offer to target certain customer groups. Some of them prepare for families with children, but they are rather in the minority. Therefore, it is a matter of individual company policy whether a hotel has some family amenities, and family tourists should check the offer of a specific facility to be sure that it meets their expectations.

General child-friendly hotel services and amenities have turned out to be the most important amenities in the opinion of parents of children in all age categories (Table 2), except for the youngest age group category (0–3 years). Krakow hotels most often meet parents’ expectations regarding this group of amenities (Table 3). This allows families to spend free time in common parts of the facility, and the children can play with their peers. The need for such amenities has also been highlighted by Brown (as cited in [21]), stressing that as many as 80% of hotel guests, when they choose a hotel, expect space for children and appropriate equipment to facilitate their play. Restaurant amenities were in second place for all parents, while in-room amenities (Table 3) were the least important. The importance of in-room amenities (including, among others, availability of baby cots, heaters and possibility of playing) in the accommodation facility has also been highlighted by Khoo-Lattmoore [20], although she has not divided amenities in her research into three groups, which the authors of this work do. The most important thing for parents of the youngest children turns out to be restaurant amenities (Table 2), such as high stools, the correct tableware and special menus. These facilities give parents of the youngest children greater freedom when eating, and children get their meal served in the correct way. As the common consumption of meals during holidays is often seen as a key element of successful holidays [82], it is not surprising that parents with young children (who have significant requirements in this respect) treat food service amenities as a priority.

An analysis of surveys conducted at hotels concerning general child-friendly hotel services and amenities shows that there are more options within the city hotel offer related to the safety and comfort of the tourist group in question (medical phone numbers, adequately wide lifts for baby carriages, access ramps) than those related to leisure and fun attractions, such as a play corner in the lobby, a play room for children or an outdoor play area. The safety at the place of stay is also highlighted in, for example, [6,21], listing it as one of the most important factors that parents take into consideration when planning travel with their children.

The presence of the latter is also subject to the situation regarding the location of city hotels within a historical city centre. Less than 25% of hotels offer special packages and discounts to families with children, which might show that city hotels focus their offer on other recipient groups. This would create a certain gap in the hotel offer, especially as parents rated this service at the highest level by allocating maximum points (5) to it.

The results of the survey related to restaurant amenities targeted at families with children are puzzling—namely, the hoteliers indicated that they have quite a large amount of tableware dedicated for children; however, their offer of children’s menus is at much lower degree. There is only a scant number of venues that offer a play corner in the restaurant.

However, taking into consideration the second stage of research, in which parents’ expectations were compared with the amenities offered by Krakow hotels, the conclusion is that the expectations of parents with children aged 0–3, 4–6, 0–6 and 4–10 years were met at a similar level for each category. In contrast, the hotels’ offer in regard to the facilities and amenities deviated from the parents’ expectations in the age groups of 7–10 and 0–10 years (Table 3), i.e., the oldest group categories, which are also the most diverse in terms of age.

At the same time, when looking at the groups of hotel amenities in various age categories of children as well as in various amenities groups, it is important to state that these were expectations of parents with the youngest children, i.e., 0–3 years, which were met to the highest degree (especially in the case of in-room amenities and restaurant amenities) (Table 3). This could show the good preparation of hotels for the accommodation of the youngest guests in the rooms to some extent, offering, e.g., cots or facilitating a peaceful meal in a hotel restaurant. As regards general child-friendly hotel services and amenities and in-room amenities, expectations regarding this group were met to the smallest degree for families with children of different ages, in the range of 0–10. In the case of services related to the restaurant amenities group, parents’ expectations in this age category of children were met to a higher degree, perhaps because of the greater number of amenities for the youngest children (these amenities’ expectations were met to a smaller degree in the group of the oldest children aged 7–10 years). As regards in-room amenities, apart from families with children of different ages, parents’ expectations were also met to a small degree in the groups of older children, namely 4–6 years and 7–10 years (Table 3), which would confirm the thesis that hotel services in rooms are best prepared to accommodate the youngest children. The above results suggest that city hotels with various family tourism amenities would meet the expectations of families with younger children and infants. This shows that the managers of the accommodation base treat children as a single, unvarying group of recipients. This corresponds to the results of [81], who stated that hotel staff often assumed that there were only two types of customers, adults and children, and that all children had the same requirements. Similar conclusions can be found in [20], where it was concluded that parents travelling with young children should be treated as a separate segment of family tourism. They should not be treated in the same way as families with older children. The need to differentiate facilities and places dedicated to children by age group was also highlighted by Buzlu and Balik [67], emphasising that it is not only about meeting different age-related needs but also about ensuring basic safety.

Conducted studies have demonstrated that some parents’ requirements regarding the services and amenities offered by hotels are similar, regardless of the hotel type. It has also been demonstrated that the facilities located in the centre of Krakow meet the expectations of parents with children at about 50% regarding general child-friendly hotel services and amenities and at about 30% regarding in-room amenities and restaurant amenities (Table 4) [7]. In a certain sense, the obtained research overlaps with other studies on the adaptation of hotel services and infrastructure to families with children, as certain expectations of parents or carers in this respect remain the same regardless of the place or time of stay. They draw attention to the fact that certain shortages and deficiencies in accommodation facilities may cause holidays or travel to be more stressful. The novelty of the research is that it addresses the topic of urban hotel offerings aimed at families with children, unlike most articles that analyse the services and offerings of holiday hotels. In the literature, generally, urban tourism is discussed in the context of adult travel and is not seen as attractive to children. Therefore, few publications discuss the adaptation of city hotels to the needs of families with children. Taking this fact into account, the results of the study may provide valuable guidance to hoteliers who wish to attract families with children to urban tourism and adapt their offer to them, thus realising the principle of inclusiveness.

## 6. Conclusions

In conclusion, it can be stated that the most important factor regarding family tourism, which should make the services and amenities offered by hotels diverse, is the age of children. It is their age that determines childrens’ needs and interests from the earliest period of infancy to early school age. The biggest challenge for the hoteliers is to prepare an offer that meets the expectations of families with children (more than 1) of different ages (the greater the age difference is, the more difficult it becomes to meet parents’ requirements regarding amenities and hotel facilities).

Many family tourism researchers confirm the thesis that children have an increasingly important influence on the direction and type of holiday and the place of accommodation, as well as activities carried out during travelling [1,2,8]. At the same time, parents are increasingly willing to organise a holiday in such a way that their children’s needs are met in the first place, because they are aware of the fact that the more children are satisfied with their holidays and the more their needs are met, the more satisfying and relaxing the holiday will be also for them [6,46,47]. This trend is also visible in the hotel offers prepared for families with children.

### Limitations

The fact that this study has been conducted in only one city can be a certain constraint to the research. We are considering carrying out similar studies in other historical cities in Poland for comparison purposes. Our work has also been carried out only in the winter months. Moreover, the hotel surveys were completed by the owners, directors and managers of the facilities, and the authors had no opportunity to see and evaluate the offered services and amenities in person. The research sample of the second phase of the study may be another limitation. The authors are aware that the results cannot be generalised to the entire population; however, they are significant enough to be a valuable indication for hoteliers who wish to incorporate the idea of inclusiveness into their offers.

## Figures and Tables

**Figure 1 ijerph-19-08321-f001:**
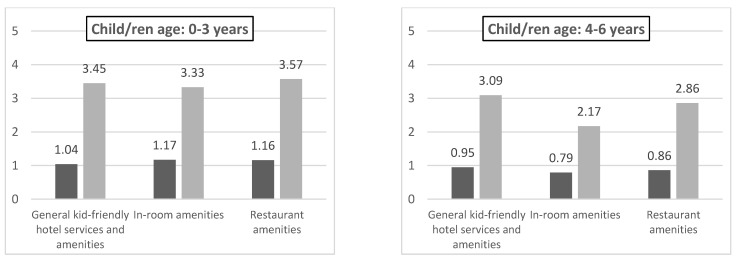
Amenities for families with children in relation to expectations.

**Table 1 ijerph-19-08321-t001:** The percentage of hotels with the amenities listed and a point value indicating parents’ expectations.

**General Child-Friendly Hotel Services and Amenities**		**PE**
Doctors’ phone numbers	53.62%	3
Discount vouchers for children’s attractions	7.25%	5
Adequately wide lifts for baby carriages	63.77%	3
A place to store baby carriages	18.84%	3
Access ramps	20.29%	4
Play area in the lobby	20.29%	3
Play room for children	7.25%	5
Gifts for kids	23.19%	1
Outdoor playground	7.25%	5
Honouring Large Family Cards	4.29%	3
Special packages and discounts for families with children	24.29%	3
**In-Room Amenities**		**PE**
Baby cots	86.96%	2
Extra beds for older children	59.42%	3
Stools for children	31.88%	2
Heaters	17.39%	3
Television with programmes for children	33.33%	3
Baby bathtubs	24.64%	1
Security (e.g., plugs, table corners, windows)	14.49%	1
**Restaurant Amenities**		**PE**
Children’s menu	24.64%	4
High stools for children	52.17%	5
Children’s tableware	69.57%	2
Play area	8.70%	4
Toilet with baby diaper changing tables	27.54%	4
A room for a mother with a baby	24.64%	3
Separate toilet for children	4.35%	1

PE—overall parents’ expectations, where 1 means “of little importance” and 5 shows a “very important” amenity. The results show some discrepancies between the services and amenities offered and the expectations of parents.

**Table 2 ijerph-19-08321-t002:** Ranking of hotel amenities based on the opinions of parents with children of different ages.

	General Child-Friendly Hotel Services and Amenities	In-Room Amenities	Restaurant Amenities
0–3 years	3.45 (3.3%)	3.33 (6.7%)	**3.57**
4–6 years	**3.09**	2.17 (29.8%)	2.86 (7.4%)
7–10 years	**3.73**	2.33 (37.5%)	3.29 (11.8%)
0–6 years	**3.82**	3.00 (21.5%)	3.43 (10.2%)
4–10 years	**3.82**	3.00 (21.5%)	3.43 (10.2%)
0–10 years	**3.64**	2.33 (36.0%)	3.14 (13.7%)
Overall	**3.59**	2.69 (25.1%)	3.29 (8.4%)

The percentages in brackets are given in relation to the top-rated group. Bold indicates the highest rated group of facilities.

**Table 3 ijerph-19-08321-t003:** Ranking of hotel amenities based on parents’ surveys compared to hotel surveys.

	General Child-Friendly Hotel Services and Amenities		In-Room Amenities		Restaurant Amenities	
	M	SD	M	SD	M	SD
0–3 years	1.04	0.61	1.17	0.67	1.16	0.66
4–6 years	0.95	0.51	0.79	0.41	0.86	0.41
7–10 years	1.05	0.65	0.77	0.47	0.86	0.47
0–6 years	1.05	0.65	1.13	0.59	1.00	0.71
4–10 years	1.05	0.65	1.13	0.59	1.00	0.71
0–10 years	0.90	0.63	0.72	0.46	1.01	0.51
Overall	1.85	1.12	0.95	0.52	1.15	0.66

**Table 4 ijerph-19-08321-t004:** The degree (expressed as a percentage) to which a hotel offer meets the expectations of parents with children of different ages.

	General Child-Friendly Hotel Services and Amenities	In-Room Amenities	Restaurant Amenities
0–3 years	30.1	35.1	32.5
4–6 years	30.7	36.4	30.1
7–10 years	28.2	33.0	26.1
0–6 years	27.5	37.7	29.2
4–10 years	27.5	37.7	29.2
0–10 years	24.7	30.9	32.2
Overall	51.5	35.3	35.0

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
