# Peer review of "Services and Amenities Offered by City Hotels within Family Tourism as One of the Factors Guaranteeing Satisfactory Leisure Time"

_ijerph, 2022, doi:10.3390/ijerph19148321_

Round 1
Reviewer 1 Report
The research has been narrowed down to one city, it is worth expanding the scope of research in the future to include other cities, for example Warsaw Poznań, etc.
it is worth conducting research in the future from the side of families using hotels to verify current research
Reviewer 2 Report
The topic of this paper is very interesting, it has a good potential to bring some novelty to this field of tourism. Methodology is properly defined and data analysis is adequate.
In conclusion and limitation section you should me more careful because the sample of your research (second phase) is a small (only 90 respondents compared to total research targeted population). So your conclusions can be only indicative and cannot be applicable to all population. Please take that in account and include it in your conclusion and limitation.
It would be also useful to include a recommendation for future research in this field.
Author Response
Please see the attachement

Reviewer 3 Report
Dear Authors,
The subject of the work is very interesting and its results will surely find a wide audience. However, due to the research period (winter months), I suggest to emphasize this issue in the discussion of the results. Perhaps, due to the weather conditions, in the summer you should expect diametrically different responses among parents.
In addition, in point 3.1, instead of describing the questions from the questionnaire - you can simply include it (survey protocol form).
There are also no characteristics of the hotels that participated in the survey, i.e. location within the city, number of rooms, types of amenities at their disposal.
Author Response
Please see the attachement

Reviewer 4 Report
The article is written in an appropriate way, but I think that the first person should be avoided. In the abstract, it is missing the methodology used and theoretical and practical implications.
In the introduction, the structure of the paper is missing.
In the literature review is missing some papers from 2021.
The final part of the data analysis is confusing. First, it is mentioned that there is no normality in the data, so you used non-parametric tests, but then you used the Student t test which is parametric. Can you use it? what is the reason? However, the data is ordinal and not normal? Very confusing. Review this part of the methodology.
The presentation of the results is confusing. It seems to me that the tables are not well referenced. See page 9. Explain further how table 3 was prepared. Page 11 t-test???
Improve the discussion, it is not clear. It seems to me that the tables are not well cited.
Conclusions are supported by results, but you should refer more, summarizing the discussion. It is missing theoretical and practical implications. This paper is interesting for the readership of this journal.
Author Response
Please see the attachement

Round 2
Reviewer 4 Report
Error in page 3 line111
Still missing references from 2021
Explain better Friedman test, what is its objective? Has it been well applied?
All the others commentaries were answered
Author Response
Please, see the attachment
